# The Clinical and Laboratory Landscape of COVID-19 During the Initial Period of the Pandemic and at the Beginning of the Omicron Era

**DOI:** 10.3390/v17040481

**Published:** 2025-03-27

**Authors:** Yulia A. Desheva, Tamara N. Shvedova, Olga S. Kopteva, Danila S. Guzenkov, Polina A. Kudar, Tatiana S. Kotomina, Daria S. Petrachkova, Elena P. Grigorieva, Anna A. Lerner, Stanislav V. Ponkratov

**Affiliations:** 1Federal State Budgetary Scientific Institution ‘Institute of Experimental Medicine’, 12, Acad. Pavlov Street, 197022 Saint-Petersburg, Russia; olga.s.kopteva@yandex.ru (O.S.K.); danila.guzenkov@yandex.ru (D.S.G.); polina6226@mail.ru (P.A.K.); tstretiak@gmail.com (T.S.K.); ya.dashook@ya.ru (D.S.P.); epgrigorieva@gmail.com (E.P.G.); 2Medical Institute, St Petersburg University, 21st Line, bldg. 8a, 199034 Saint-Petersburg, Russia; 3Vsevolozhsk Clinical Interdistrict Hospital, 20, Koltushskoe Highway, Leningrad Region, 188643 Arkhangelsk, Russia; toma_nn@mail.ru (T.N.S.); ponkratovsv@vkmb.ru (S.V.P.); 4Department of Clinical Laboratory Diagnostics, Biological and General Chemistry, Federal State Budgetary Educational Institution of Higher Education North-West State Medical University, 191015 Saint-Petersburg, Russia; sever67@bk.ru

**Keywords:** SARS-CoV-2, Omicron, HRM analysis, N protein, COVID-19, complement C3, influenza coinfections

## Abstract

Introduction: Severe acute respiratory syndrome coronavirus 2 (SARS-CoV-2) underwent significant mutations, resulting in the Omicron variant. Methods: In this study, we analyzed blood samples from 98 patients with acute coronavirus disease 19 (COVID-19) hospitalized during the initial SARS-CoV-2 wave and the onset of Omicron in 2021. High-resolution melting (HRM) analysis of PCR products was used to analyze RNA extracted from clinical samples collected in July and November 2021 from patients infected with SARS-CoV-2. Results: HRM analysis revealed a characteristic deletion in the N protein RNA of the virus isolated in November 2021, associated with the Omicron variant. Elevated levels of inflammatory markers and interleukin-6 (IL-6) were observed in both waves of COVID-19. Complement levels and IgG and IgM antibodies to SARS-CoV-2 were detected more often during the second wave. An increase in hemagglutinin-inhibiting (HI) antibodies against influenza viruses was observed in paired blood specimens from moderate to severe COVID-19 patients during both outbreaks. Conclusions: Patients admitted during both waves of COVID-19 showed a significant rise in inflammatory markers, suggesting that Omicron triggers inflammatory responses. The rapid formation of IgM and IgG in Omicron may indicate a faster immune response. Seasonal flu may negatively impact the clinical course of coronavirus infections.

## 1. Introduction

The clinical and laboratory landscape of COVID-19 has shifted significantly over the course of the pandemic [1]. Mutations in the SARS-CoV-2 virus have led to the emergence of various antigenic variants, including the EG.5 lineage (Eris), the descendant of the XBB.1.9.2 lineage, as well as XBB.1.16 (Arcturus), XBB.1.5 (Kraken), and other variants of the Omicron coronavirus [2]. The new variant of the coronavirus, B.1.1.529, named ‘Omicron’, was discovered on 24 November 2021 in South Africa from a patient’s specimen sample that was collected on 9 November 2021 [3]. Shortly after its identification, Omicron was detected in several other countries, marking its rapid global spread [4]. In Russia, the Omicron variant began to spread rapidly in December 2021, and it currently completely dominates the territory of Russia (100% of all samples studied) [5].

With the advent of Omicron, the speed of spread of coronavirus has increased and the incubation period has decreased [6]. Omicron contains more than 30 mutations in its spike protein, which is significantly higher than in previous variants [7]. Many of these mutations help Omicron evade the immune response generated by previous infections and vaccinations [8].

The main danger of this new subtype of the virus is that it can evade the body’s immune system [9,10]. Before Omicron, natural immunity provided robust and lasting protection against reinfection [11]. However, in the Omicron period, protective immunity was only observed among recently recovered people and declined rapidly over the course of a year. This emphasizes the need for continuous monitoring of the virus and its mutations, as well as regular updates to SARS-CoV-2 vaccines to restore immunity and combat ongoing viral immune evasion [11].

Despite the detected decrease in neutralizing antibody responses to Omicron, vaccines continue to provide significant protection against severe illness [12]. Epidemiological studies have shown that Omicron infections may result in a variety of symptoms, often associated with mild forms of the illness, especially in vaccinated individuals [10,13]. For all ages, the risk of COVID-19-related death, regardless of the number of comorbid conditions, is lower with Omicron than with Delta [14].

Although the new strains have a milder course of the disease, they can still cause serious complications [15]. Omicron has demonstrated the highest transmission rate among other variants of SARS-CoV-2 [4], leading to significant increases in cases, especially among unvaccinated people or those with pre-existing health conditions [16].

While the S protein of Omicron undergoes substantial changes, promoting transmission and immune evasion, the changes in the N protein are more modest, primarily relevant to diagnostic considerations and antibody tests [17]. The N protein is targeted by the host’s immune response, regulating innate immune responses such as type I interferon (IFN-I) signaling and cytokine production [18]. Antibodies produced against the N proteins can be part of an adaptive immune response [19]. Like other regions of viral genomes, the N gene can accumulate mutations over time, affecting protein structure and function. These changes can lead to alterations in viral fitness and virulence, as well as the ability to avoid the immune response [20]. Some SARS-CoV variants may develop mutations in their N genes that allow them to evade detection by the immune system and impact vaccine effectiveness and natural immunity [21]. Studying evolutionary changes in SARS-CoV not only in S proteins but also in N proteins provides a more complete understanding of its biology, evolution, and interaction with the human immune system.

Since the beginning of the SARS-CoV-2 spread, there have been several reports of co-infection with influenza and coronavirus infections [22]. Since the beginning of the spread of the SARS-CoV-2 virus, there have been quite a few reports of coinfection with influenza viruses and coronavirus infection [13,14,15,16,17,18,19,20,21,22,23,24,25]. During quarantine and due to the widespread use of personal protective equipment, influenza virus circulation has significantly decreased by 2022. Even the influenza B/Yamagata virus disappeared from circulation for the first time in 40 years [26]. Under such conditions, population immunity against influenza decreases, and the likelihood of the emergence of new influenza virus antigenic variants increases.

The purpose of this study is to compare clinical features, inflammatory markers, and adaptive immune parameters in patients with COVID-19 who were hospitalized during the first wave of the pandemic and at the end of 2021 (beginning of the ‘Omicron’ era). One of the objectives was to evaluate the contribution of co-infections with influenza to the severity of COVID-19 during these two periods.

## 2. Materials and Methods

### 2.1. Ethics Statement

All study participants signed a written informed consent. The study was approved by the Local Ethics Committee of the Federal State Budgetary Scientific Institution ‘IEM’ (protocol 1/23 dated 20 April 2023).

### 2.2. Study Participants and Samples

In this retrospective cohort study, a total of 45 sera from patients with acute COVID-19 obtained at the beginning of the SARS-CoV-2 outbreak (including 28 paired samples) and 53 sera from patients hospitalized at the end of 2021 (including 14 paired samples) were studied. The samples were provided by the Vsevolozhsk Clinical Interdistrict Hospital, Leningrad Region, Russian Federation. The cohort was divided into 3 groups according to the severity of the disease at hospital admission. The severity was assessed according to the interim recommendations for the prevention, diagnosis, and treatment of coronavirus disease 2019 (COVID-19), version 8. Mild disease is characterized by body temperature < 38 °C, coughing, weakness, and sore throat without criteria for moderate or severe disease. Moderate disease is characterized by body temperature > 38 °C, respiratory rate > 22/min, dyspnea on exertion (shortness of breath), radiographic changes typical of viral infection (minimal or moderate lesion volume), oxygen saturation < 95%, and serum C-reactive protein (CRP) > 10 mg/L. Severe disease is characterized by respiratory rate > 30/min, oxygen saturation < 93%, unstable hemodynamics (systolic blood pressure less than 90 mm Hg or diastolic blood pressure below 60 mm Hg, diuresis below 20 mL/h), lung changes on radiograph typical of viral infection (significant or subtotal lesion volume, >75%), and arterial lactate > 2 mmol/L. Extremely serious diseases include acute respiratory distress syndrome (ARDS). Nasopharyngeal and pharyngeal swabs were tested for real-time PCR upon hospital admission.

### 2.3. Molecular Genetic Analysis

The nucleotide sequence analysis was performed in the Ugene program. Phylogenetic trees were constructed using the MEGA11 statistical method maximum likelihood [27]. Alignment was performed according to the reference genome of Wuhan-Hu-1 (NC_045512.2). The sequences of the full-length N protein of the SARS-CoV-2 virus were used as follows: BS 016103.1, ON 442223.1, OK 501819.1, PQ 169822.1, OR 275853.1, OR 818056.1, MZ 140764.1, OR 729868.1, PQ 281369.1, PQ 370471.1, OR 457591.1, BS 014763.1, PQ 389522.1, OQ 852551.1, OQ 331796.1, PP215584.1, PQ102435.1, OR958502.1, PQ 515554.1, PP 103560.1, OQ 050741.1, PP 604035.1, OM 247072.1, OP 855515.1, OK 194572.1, LC 573289.2, PQ 437118.1, PQ 210560.1, PQ395243.1, PQ047794.1, PQ508588.1, PQ358595.1, and PQ481382.1.

Primers for the N gene study by high-resolution melting curve analysis (HRM analysis) and sequencing were designed using the Primer3Plus online server (Table 1).

Primers were synthesized by Alcor-Bio (St. Petersburg, Russia). We performed a molecular genetic analysis of SARS-CoV-2 RNA from two nasopharyngeal and pharyngeal swabs from patients with acute COVID-19, obtained in July 2021 (swab 1) and November 2021 (swab 2). The reference RNA for the Wuhan variant of the SARS-CoV-2 virus, kindly provided by the Institute of Influenza, was used as the control sample.

Reverse transcription PCR was performed using Moloney Murine Leukemia Virus Reverse Transcriptase (M-MulV RT), 200,000 units/mL; RNase inhibitor, 40 units/µL; 10× buffer; deoxyribonucleoside triphosphates; and SsoFastEvaGreen mixtures on a CF X96 (Biorad, Hercules, CA, USA) device. Melting curves were analyzed using the Precision Melt analysis software (version 1.3) as described previously [28].

### 2.4. Laboratory Data

Serum C-reactive protein (CRP) and fibrinogen concentrations were determined by the turbidimetric method using BioSystem reagents (BioSystems, Barcelona, Spain). Interleukin-6 (IL-6), tumor necrosis factor alpha (TNF-α), or interferon 1 alpha (IFN-α) were determined using ELISAs (Vector-best, Novosibirsk, Russia) according to the manufacturer’s instructions. Serum complement C3 levels were measured using an ELISA kit (Cloud-Clone Corp., Wuhan, China) according to the manufacturer’s instructions. All blood samples were tested without heating. The neutrophil/lymphocyte ratio (NLR) was calculated based on clinical data from blood tests using the formula for absolute neutrophils/absolute lymphocytes.

### 2.5. Hemagglutination Inhibition Test (HI)

To determine the level of HI antibodies to influenza viruses in the blood samples studied, the influenza viruses obtained from the Department of Virology of the Institute of Experimental Medicine were used as follows: A/New York/61/2015(H1N1)pdm09, A/Hong Kong/4801/2014 (H3N2), B/Colorado/06/2017 (B/Victoria/2/87 lineage) (2020), and A/Guangdong-Maonan/SWL1536/2019(H1N1)pdm09, B/Austria/06/2017 (B/Victoria/2/87 lineage) (2021). Blood sera were pretreated with an enzyme-destroying enzyme, RDE (Denka Seiken, Tokyo, Japan), as described previously [29]. RDE-treated and 1:10 diluted blood sera were titrated in short rows on a 96-well U-bottomed immunoassay plate (’Medpolymer’, St. Petersburg, Russia) to obtain a series of 2-fold dilutions (in 25 μL of PBS): 1:10, 1:20, 1:40, and so on. Then, standard doses of virus (8 agglutinating units, AU) in a volume of 25 μL were added to each well. After 30 min at room temperature, 50 μL of 0.75% suspension of human red blood cells (RBCs) of group I(0) were added and kept for another 40 min under similar conditions for RBC sedimentation. The serum titer was determined as the reciprocal of the dilution of the last well in which no hemagglutination occurred. A fourfold or greater increase in the level of HI antibodies was considered reliable seroconversion.

### 2.6. Statistical Analysis

Statistical data processing was carried out using the Prism 8 software package (GraphPad software, v10.4.0, San Diego, CA, USA). Medians (Me) and lower and upper quartiles (Q1; Q3) were calculated and used to present the antibody response and blood test levels. In cases with normal distribution of the studied characteristics, M ± σ, with M being the mean value and σ being the standard deviation. Comparisons between independent groups were made using non-parametric tests: for multiple comparisons, Friedman’s test (F-test (ANOVA) or Kruskal–Wallis (Kruskal–Wallis ANOVA) and Mann–Whitney (Mann–Whitney U test) to assess intragroup differences. A *p*-value of <0.05 is considered statistically significant.

## 3. Results

### 3.1. Molecular Genetic Analysis of the N Protein Gene of SARS-CoV-2 Antigenic Variants

A phylogenetic tree of the N protein of the most representative SARS-CoV-2 variants was constructed in comparison with the original ‘Wuhan’ variant (Figure 1).

We performed a molecular genetic analysis on N protein RNA from two nasopharyngeal samples from patients with acute COVID-19, obtained in July 2021 (swab 1) and November 2021 (swab 2). The Wuhan reference RNA from the SARS-CoV-2 virus variant was used as a control. The graphs showed a clear difference in the melting patterns for the fragments of swabs 1 and 2 and the reference RNA (Figure 2a,b). In a swab obtained in November 2021, a characteristic deletion of subvariants from the Omicron lineage was detected (Figure 2a). In a multiple alignment analysis of the major SARS-CoV-2 variants, we found that this deletion was also detected in other Omicron variants (Figure 2b).

When analyzing the fragment flanked by primers 107–202, a single nucleotide substitution 188A → G was identified in swab 1, leading to an amino acid substitution D63G, characteristic of the Delta strain variants B.1.617.2 and AY.9.2 [32] (Figure 2c). Wash 2, as shown by sequencing, has a previously undescribed missense mutation 170C → T in this region, leading to an amino acid substitution T57I (Figure 2c).

The multiple alignment of the nucleocapsid proteins of SARS-CoV-2 virus strains and nasopharyngeal swab sequences from patients showed the divergence of the Omicron strain during the evolution of the new coronavirus infection (Figure 2d). It was shown that both swabs differ from the original Wuhan variant, while swab 1 belongs to the same phylogenetic branch as the Delta variant and wash 2 is closer to the Omicron variant (Figure 2d). Thus, it was shown that by the end of 2021, the SARS-CoV-2 virus had acquired the characteristics of Omicron, which had been officially registered in Russia since December 2021 [2].

### 3.2. The Main Data on the Observed Patient Cohorts

Table 2 presents the main characteristics of acute COVID-19 patients examined in March 2020 and November 2021. The main differences between the two groups were as follows: the median time from the onset of symptoms to hospitalization was shorter for the second group (*p* = 0.01); the proportion of positive PCR test results for SARS-CoV-2 was greater in the second group than in the first one; in 2021, almost 70 percent of subjects received two doses of the S protein-based vaccine SARS-CoV-2 (Sputnik-V), unlike the patients in the first cohort, which were examined prior to the introduction of anti-COVID vaccination. Accordingly, the proportions of individuals possessing serum IgG and IgM antibodies to the SARS-CoV-2 virus were statistically significantly higher in the second cohort of those examined. Additionally, viral RNA was not detected in the blood of patients from the second cohort, in contrast to the first cohort, and there were no fatalities in the second wave, whereas 24.1 percent of cases resulted in death in the first cohort (Table 2). At the same time, there were no significant differences between the two groups in terms of such parameters as frequency of concomitant conditions and bacterial co-infections, as well as the age and gender distribution of the studied groups (Table 2).

There were no significant differences in the main parameters between men and women in the examined cohorts (Appendix A), and also there were almost no statistically significant differences between the parameters studied depending on the age of patients (Appendix A). The exception was that, in the second cohort, the levels of TNF-α and IFN-α were statistically significantly lower in patients younger than 65 compared to older ones (*p* = 0.04 and *p* = 0.001, respectively, Appendix A). There was no significant difference between vaccination statuses among patients examined in 2021 (Appendix A).

### 3.3. The Levels of Inflammatory Markers and Cytokines in the Serum Samples of Patients Examined

Figure 3 shows the results of a study on the content of inflammatory markers in hospitalized patients with acute COVID-19 during the two waves of the disease. The levels of CRP and NLR did not differ statistically significantly between the two cohorts examined. At the same time, a statistically significant decrease in fibrinogen was recorded in hospitalized COVID-19 patients during the second wave (Figure 3).

The mean levels of complement C3 were higher in the second cohort (Figure 3).

Considering that CRP and NLR levels were elevated in patients not only during the first wave of COVID-19 infection but also in late 2021, we analyzed these markers among patients with different degrees of severity (Figure 4a,b). It was found that in 2021, patients with moderate to severe COVID-19 disease had mean CRP and NLR levels that were also significantly higher compared to those with mild disease (Figure 4b).

TNF-α and IFN-α levels were significantly lower among patients from the second cohort (Figure 5). However, IL-6 did not differ significantly between the two groups, and in most patients in the second group, it still significantly exceeded the reference levels (Figure 5).

### 3.4. Antibodies to SARS-CoV-2 Analyzed Patient Cohorts

IgG seropositive rates were higher in group 1, while IgM levels were higher in group 2 (Figure 6).

### 3.5. Increases in Serum HI Antibodies to Influenza Viruses in Paired Blood Sera

Fourfold or more increases in HI antibody levels to influenza viruses in pairs of serum samples were determined, as there was previous evidence of coinfection with these viruses in acute COVID-19. The main characteristics of the examined patients are presented in Table 3.

It was shown that in both moderate and severe COVID-19 cases, fourfold or more increases in HI antibodies to influenza A and B viruses were observed in both cohorts (Table 3). In groups with milder COVID-19, seroconversion to influenza viruses was not detected. In 2020 and 2021, most of the increases in antibody levels were seen for the influenza A/H1N1pdm09 and B/Victoria viruses, respectively. (Figure 7).

## 4. Discussion

Analysis of the genomic diversity of SARS-CoV-2 revealed several stages in the spread of the most representative strains of the virus during the pandemic [16]. Periods with homogeneity of the pathogen population, where the Alpha, Delta, or Omicron variants dominated, were followed by more complex stages involving the simultaneous circulation of several strains. This was due to the adaptation of the virus. The Alpha variant, with its high transmissibility and virulence at the beginning of the pandemic, contributed to the emergence of Delta, which caused a sharp increase in the incidence rate. Active vaccination in Russia began in 2021 and led to a rise in seroprevalence up to 50% [2,5]. Gradually, Delta adapted to a greater proportion of people who were not susceptible, leading to the formation of Omicron. Delta-Omicron recombinant strains were also found, with the N-terminal portion of the N gene being identical to that of the Omicron sequence [17]. We used HRM analysis to compare two SARS-CoV-2 viruses isolated in 2021.

We used HRM-analysis to determine differences between the N protein RNA of two SARS-CoV-2 isolates obtained during 2021. The isolate obtained in late 2021 acquired Omicron features, as a deletion of nucleotides 90–98 was detected, which leads, as previously shown, to the loss of ERS31 amino acids without a frameshift [32]. Given the low cost and ease of implementation, HRM analysis is suitable for rapid screening of large numbers of isolates for genotyping purposes.

The study of inflammation markers and cytokines in relation to Omicron and other SARS-CoV-2 variants is an important area of research that can help us understand viral pathogenicity, influence clinical practice and public health policies, as well as therapeutic development. Unlike the original Wuhan strain, Omicron tends to cause infections in the upper airway rather than deeper lung tissues. This could be for a number of reasons, including changes in receptor binding due to a large number of mutations in the S protein and potential utilization of alternative viral entry pathways [13]. This may reduce the impact on critical organs like the lungs, leading to less severe respiratory issues and lower systemic inflammation. Also, a decrease in inflammatory markers in Omicron may be associated with vaccination. Thus, in our study, about 70% of the patients examined in late 2021 were vaccinated against COVID-19, and, in general, despite the presence of severe cases, no mortality was observed. Indeed, the proportion of seropositive patients is significantly higher in the second cohort compared with the first cohort (Table 2). In 2021, no viremia was recorded using the detection of SARS-CoV-2 virus RNA in the blood of patients, and some blood parameters were lower compared to hospitalized patients in 2020. Thus, fibrinogen levels as well as TNF-α and IFN-α levels were statistically significantly lower in patients from the second cohort. It has been shown previously that elevated fibrinogen levels can serve as a marker of inflammation and severity of COVID-19 [33]. Our study showed that, in all patients examined during the first wave of COVID-19, fibrinogen levels were elevated. During Omicron in 2021, some hospitalized patients had fibrinogen levels within reference values, and the average level was significantly lower than in 2020 (Figure 3). The reduction in TNF-α and IFN-α levels in the second cohort shown in our study may reflect a reduction in the overall burden on the immune system with Omicron. However, in hospitalized patients, Omicron still may cause significant systemic inflammation. And indeed, the levels of CRP, NLR, and IL-6 did not differ significantly between the two groups of patients and remained significantly above reference values in both the 2020 and 2021 groups. In 2021, individuals with moderate to severe COVID-19 experienced elevated levels of CRP and NLR compared to those with mild COVID-19 (Figure 4), just like during the first wave of the disease. The observation that certain pro-inflammatory markers become lower in hospitalized patients during the Omicron wave, while IL-6 remains high, reflects a combination of reduced viral pathogenicity, differences in immune responses, and the distinct roles of specific cytokines in inflammation.

In addition, among patients from the second cohort, the level of serum complement C3 was significantly higher than that in patients examined in 2020 (Figure 3). Complement component C3 is a protein in the complement system, which is part of the body’s immune response. In the context of COVID-19, complement activation has been observed as a significant part of the immune response to the virus [34]. The population’s level of immunity due to vaccination or prior infection can affect the immune response to a new variant. Omicron’s impact on serum complement levels may also reflect differences in how people with pre-existing immunity respond compared to those who were infected during the first wave. The increased complement C3 content in 2021 compared to 2020 may confirm the activation of the complement system in response to breakthrough infection with Omicron. This evasion could lead to a more pronounced reliance on the complement system as the immune system attempts to mount a defense [35]. Nevertheless, some studies suggest that complementary dysregulation, including the involvement of C3, may play a role in the development of long COVID. Previously, it was shown that activation of the complement system leading to the formation of the membrane attack complex plays a role in COVID-19 pathogenesis [36]. This can lead to the release of self-antigens from damaged tissues and trigger an autoimmune response, as the immune system begins to target these self-antigens, leading to tissue inflammation and symptomatology associated with autoimmunity [36,37,38]. Persistent complement activation could contribute to ongoing inflammation and immune dysregulation that characterizes long COVID symptoms. Markers of complement activation, including elevated levels of C3, may provide insights into the underlying mechanisms contributing to prolonged COVID-19 symptoms.

A separate issue is the aggravation of the clinical picture of COVID-19 due to coinfection with influenza viruses. In this regard, there was great similarity between the two examined cohorts. Since a 4-fold increase in HI antibodies is the ‘gold standard’ of seroconversion to influenza [29], the data obtained suggest that in moderate cases of COVID-19, coinfection with influenza viruses could possibly be present. The fact that in 2020 there were increases in the influenza A/H1N1pdm09 virus and in 2021 in the B/Victoria virus may coincide with the pattern of circulation of influenza viruses in the indicated epidemic seasons [39]. The problem of influenza and other respiratory infections in COVID-19 has been repeatedly discussed in the scientific literature. The fact that the incidence of seasonal influenza often coincides with an increase in the incidence of COVID-19 may negatively contribute to the clinical picture of coronavirus infection [40,41].

This indicates the need for increased attention to influenza vaccination in the context of preventing severe forms of coronavirus infection. Thus, it has been shown that vaccination with a quadrivalent influenza vaccine not only prevents severe cases of COVID-19 but also helps to ’pre-stimulate’ the immune system for faster and earlier responses to the pathogen [42].

## 5. Conclusions

Thus, the analysis of clinical isolates of the SARS-CoV-2 virus using HRM analysis showed the deletion characteristic of the Omicron variant in the N protein RNA at the end of 2021. HRM analysis allows for the differentiation of changes in the antigenic structure of SARS-CoV variants.

Patients hospitalized both in the first wave of the disease and in 2021 showed a significant increase in CRP, NLR, and IL-6. The data that IL-6 and inflammatory markers are significantly increased in Omicron and average complement levels are higher in 2021 compared to the first wave can be assumed to be the fact that the high transmission capacity of new coronavirus variants may lead to the fact that severe consequences may be observed in a larger number of people.

IgG and IgM to SARS-CoV-2 were detected statistically significantly more often at the beginning of hospitalization during the second wave of COVID-19, which may indicate a more rapid immune response due to the formation of memory B cells after previous infections or immunizations.

And finally, the fact that seroconversions to influenza viruses were detected in moderate and severe cases of COVID-19 indicates an additional negative contribution of influenza to the pathogenesis of coronavirus infection.

Further studies are needed to determine whether clinical presentation and blood parameters change in response to changes in the SARS-CoV-2 virus in breakthrough infections and in patients with post-COVID syndrome.

The main limitation of this study is the small number of samples in the molecular study. HRM analysis includes only two samples from different periods, which may not be representative of the full range of SARS-CoV-2 variants present in the population at a given time.

## Figures and Tables

**Figure 1 viruses-17-00481-f001:**
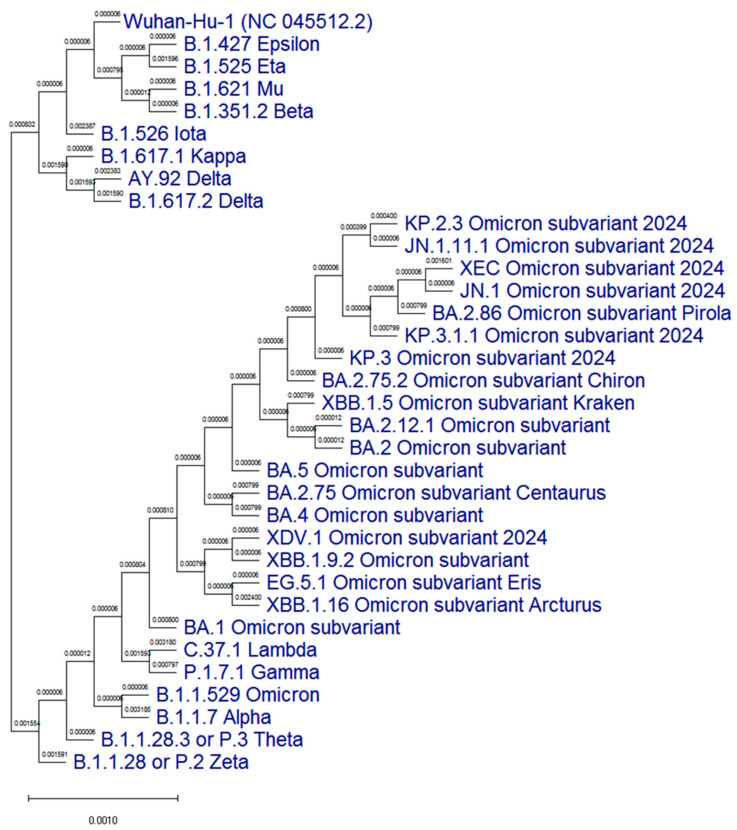
Phylogenetic analysis of the N protein (1–419 amino acids) of the 33 most representative SARS-CoV-2 variants, including the Wuhan-Hu-1 reference genome. Multiple alignment was performed with the ClustalW method [30]. The branch length represents the number of substitutions per site. The names of the compared genomes of SARS-CoV-2 variants according to the Pango nomenclature [31] are shown in blue.

**Figure 2 viruses-17-00481-f002:**
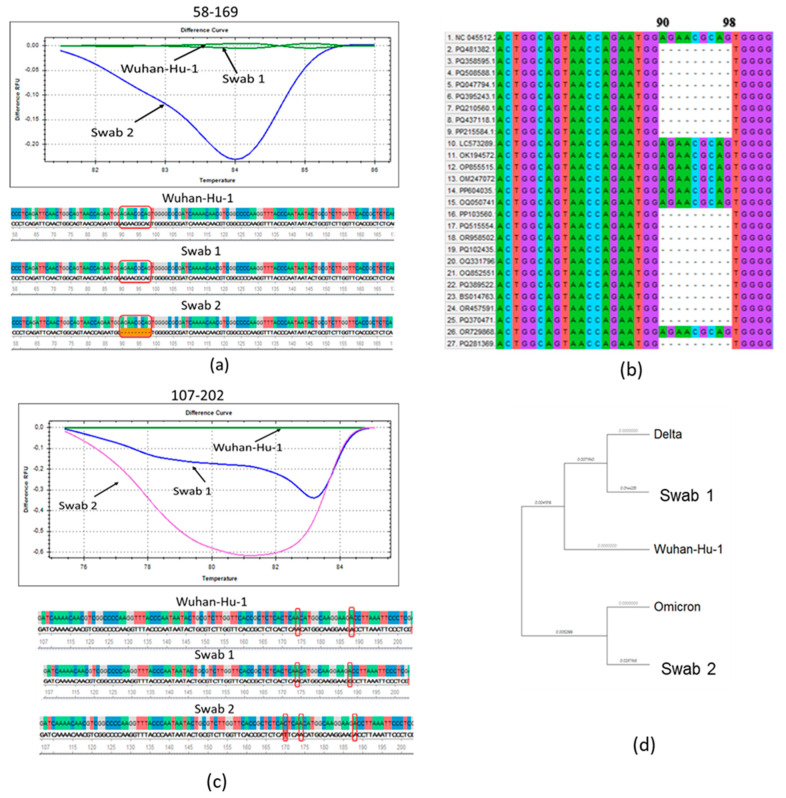
Molecular genetic analysis of the N protein fragments. In the consensus nucleotide sequence, individual nucleotides are highlighted in different colors. Differences between the original Wuhan variant and the analyzed clinical samples are highlighted in red frames. (**a**) High-resolution melting curves and nucleotide sequence analysis using the F58-R169 primer set. (**b**) High-resolution melting curves and nucleotide sequence analysis using the F107-R202 primer set. (**c**) Multiple alignment in the region of the deletion characteristic of subvariants of the Omicron strain, determined by the primer pair F58-R169. (**d**) Phylogenetic analysis of the SARS-CoV-2 N protein (1–141 aa) from the Wuhan-Hu-1 reference sequence, as well as two nasopharyngeal washes and Omicron BA.1 and Delta B.1.617.2 strains. MEGA11 was used for phylogenetic analysis, with maximum likelihood statistical methods. Branch lengths represent the number of nucleotide substitutions per site.

**Figure 3 viruses-17-00481-f003:**
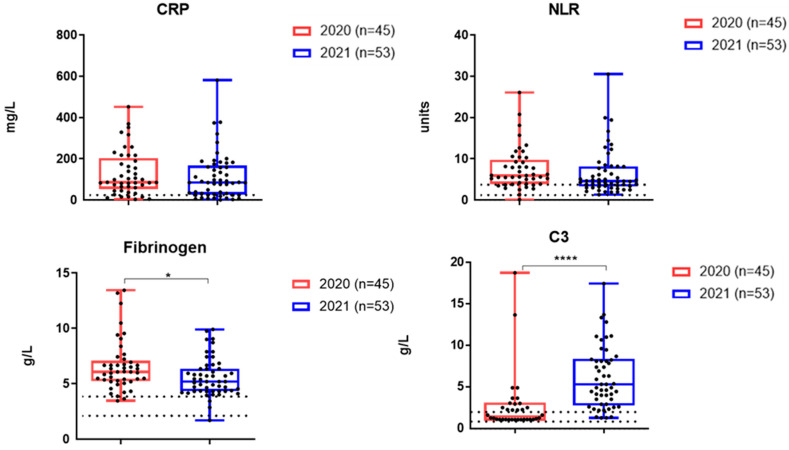
The mean levels of inflammatory markers in the blood serum of patients with acute COVID-19 during days 1–3 of hospitalization. Patients were admitted to the hospital in March 2020 and November 2021. *—*p* < 0.05; ****—*p* < 0.0001.

**Figure 4 viruses-17-00481-f004:**
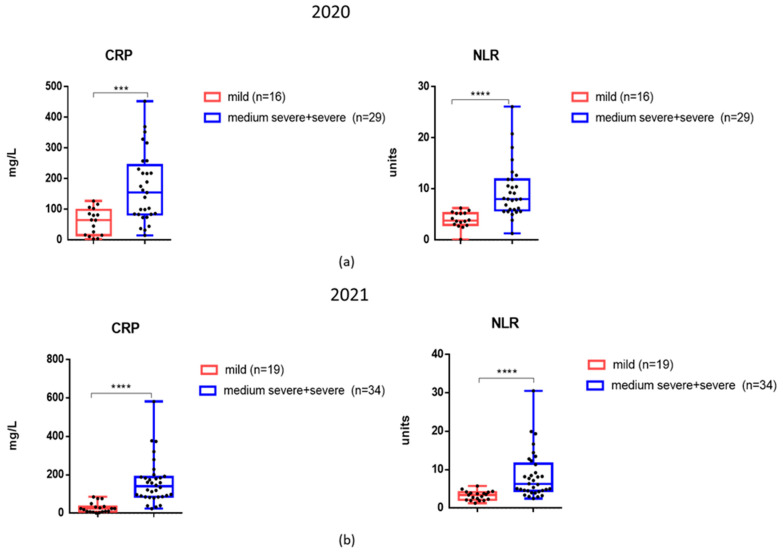
The mean levels of inflammatory markers in the blood serum of acute COVID-19 of varying severity during days 1–3 of hospitalization. (**a**) Patients were admitted to the hospital in March 2020. (**b**) Patients were admitted to the hospital in November 2021. ***—*p* < 0.001; ****—*p* < 0.0001.

**Figure 5 viruses-17-00481-f005:**
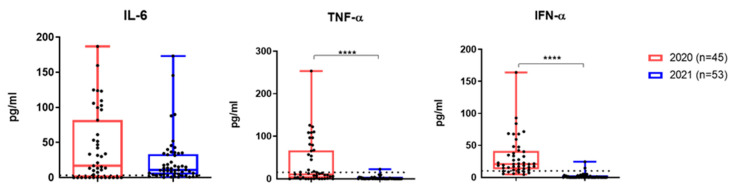
Results of the study of the content of early cytokines and type I interferon in the blood of patients with acute COVID-19 during the first 3 days of hospitalization. Patients were admitted to the hospital in March 2020 and November 2021. Reference values for IL-6: 1.3–6.8 pg/mL; for TNF-α: 0–8.21 pg/mL; for IFN-α: <10 pg/mL. ****—*p* <0.0001.

**Figure 6 viruses-17-00481-f006:**
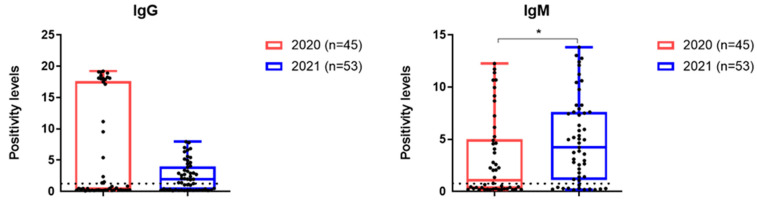
Results of a study on the content of antibodies specific to the SARS-CoV-2 virus in the blood serum of patients with acute COVID-19 during the first three days of hospitalization. Patients were admitted to the hospital in March 2020 and November 2021. *—*p* < 0.05.

**Figure 7 viruses-17-00481-f007:**
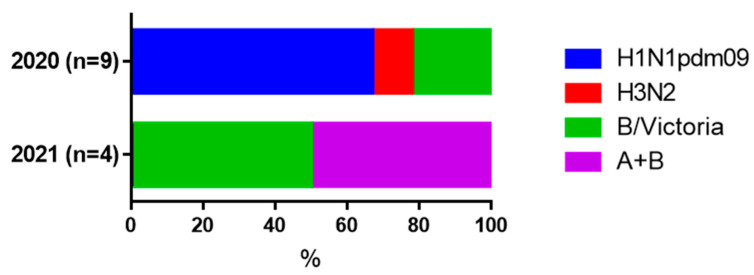
Proportions of a four-fold and more increase in HI antibodies to the influenza virus among all detected seroconversions.

**Table 1 viruses-17-00481-t001:** Primers designed for the analysis of the N protein gene of the SARS-CoV-2 virus.

Position in the Nucleotide Chain	Nucleotide Sequence (5′-3′)	Length of Fragment, bp	T Annealing, °C
F58	CCCTCAGATTCAACTGGCAGT	112	
R169	TGAAGAGCGGTGAACCAAGAC		55

**Table 2 viruses-17-00481-t002:** Characteristics of the study patients.

Characteristic	1st Cohort (2020), *n* = 45	2nd Cohort (2021), *n* = 53	*p*=
Age, Me (Q1; Q3)	62.00 (55.00; 70.00)	67.00 (56.50; 76.00)	0.15
Males	22 (48.9%)	28 (52.8%)	0.5
Females	23 (51.1%)	25 (47.2%)	0.5
<65 years old	28 (62.2%)	23 (43.4%)	0.43
≥65 years old	17 (37.8%)	30 (56.6%)	0.43
Mild COVID-19	16 (35.6%)	19 (35.8%)	0.57
Medium severe + severe COVID-19	29 (64.4%)	34 (64.2%)	0.57
Days from onset of illness, Me (Q1; Q3)	7.00 (5.00; 9.00)	4.00 (3.00; 5.00)	<0.0001
Positive PCR test for SARS-CoV-2 on the day of hospitalization	22 (48.9%)	53 (100%)	<0.001
Viremia (positive serum PCR-test)	12 (26.7%)	0 (0%)	<0.0001
SARS-CoV-2 vaccination	0 (0%)	37 (69.8%)	<0.001
Comorbidities:			
Cardiovascular	31 (68.8%)	31 (58.5%)	0.2
Diabetes	9 (20%)	6 (11.3%)	0.17
Chronic pulmonary disorders	5 (11.1%)	3 (5.6%)	0.27
Bacterial coinfections	18 (40.0%)	22 (41.5%)	0.52
Lethal outcome	13 (24.1%)	0 (0%)	<0.001
Positive for IgG	15 (33.3%)	31 (58.5)	0.01
Positive for IgM	15 (33.3%)	40 (45.5%)	<0.0001
Influenza seroconversions among paired samples	9 out of 28 (32.2%)	5 out of 14 (35.7%)	0.12

**Table 3 viruses-17-00481-t003:** Analysis of paired serum samples to detect four-fold increases in antibodies to influenza virus.

Characteristic	1st Cohort (2020)	2nd Cohort (2021)	*p*=
Number of patients	28	14	
Mild	9 (32.1%)	5 (35.7%)	0.54
Medium severe + severe	19 (67.9%)	9 (64%)	0.54
Average period between the 1st and 2nd serum	4 (3.0; 4.0)	4 (3.0; 5.0)	0.18
Seroconversion to influenza in mild COVID-19	0 (0%)	0 (0%)	n/a
Seroconversion to influenza in medium severe + severe COVID-19	9 (47.4%) ^1^	4 (44.4%)	0.69
Seroconversion to influenza viruses (HI)			
A/H1N1pdm09	6 (21.4%)	1 (7.1%)	0.19
A/H3N2	1 (3.6%)	1 (7.1%)	0.67
B/Victoria	2 (7.1%)	4 (28.6%)	0.08

^1^*p* = 0.01 compared to mild COVID-19.

## Data Availability

All data are contained in the text of the paper and in the Appendix A.

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
