# Peer review of "The Clinical and Laboratory Landscape of COVID-19 During the Initial Period of the Pandemic and at the Beginning of the Omicron Era"

_viruses, 2025, doi:10.3390/v17040481_

Round 1
Reviewer 1 Report
Comments and Suggestions for Authors
Interesting paper concerning the changes in laboratory data connected with virus genetic variants during SARS-CoV-2 pandemic progression. Some comments:
1) No data on anti-SARS-CoV-2 antibody detection methods. Were antibodies to S and N protein measured separately? (only cumulative data given).
2) No data given wjth what vaccines studied patients were vaccinated. Were vaccines to N-protein or to S-protein (Sputnik) used? This could give more detailed data on antibody levels in patient's sera. Is to piossible to specify these data?
3) Original data obtained in this study confirmed that inflammatory markers levels were lower in patients infected with Omicron strain compared to initial Wuhan strain. These findings can explain more mild clinical features in these patients. Probably it may be discussed more clear.
Author Response
Interesting paper concerning the changes in laboratory data connected with virus genetic variants during SARS-CoV-2 pandemic progression.
Response. We thank the reviewer for the positive assessment of our work.
Some comments:
- No data on anti-SARS-CoV-2 antibody detection methods. Were antibodies to S and N protein measured separately? (only cumulative data given).
Response: We thank the reviewer for this question. Antibodies to the SARS-COV-2 virus were measured using commercial test systems designed for current serological and epidemiological diagnostics. The manufacturer does not indicate which of the coronavirus antigens the antibodies are detected to, as this is a commercial secret. We can only assume that they are to the whole SARS-COV-2 virus.
- No data given wjth what vaccines studied patients were vaccinated. Were vaccines to N-protein or to S-protein (Sputnik) used? This could give more detailed data on antibody levels in patient's sera. Is to piossible to specify these data?
Response: We thank the reviewer for this question. All patients received 2 doses of the S-protein-based vaccine SARS-CoV-2 (Sputnik-V). We indicated in the text.
- Original data obtained in this study confirmed that inflammatory markers levels were lower in patients infected with Omicron strain compared to initial Wuhan strain. These findings can explain more mild clinical features in these patients. Probably it may be discussed more clear.
Response: We thank the reviewer for this question.
We added to the Discussion:
“Unlike the original Wuhan strain, Omicron tends to cause infections in the upper airway rather than deeper lung tissues. This could be for a number of reasons, including changes in receptor binding due to a large number of mutations in the S protein and potential utilization of alternative viral entry pathways [13]. This may reduce the impact on critical organs like the lungs, leading to less severe respiratory issues and lower systemic inflammation”.
Reviewer 2 Report
Comments and Suggestions for Authors
The work by Yu. A. Desheva et al. entitled "The clinical and laboratory landscape of COVID-19 during the initial period of the pandemic and at the beginning of the Omicron era" is an interesting piece of literature. The article presents a retrospective clinical and laboratory analysis of hospitalized patients with COVID-19 in two different periods of the pandemic - the initial wave and at the turn of 2021, when the Omicron variant appeared. The authors examine differences in inflammatory biomarkers, immune response, and the impact of co-infection with influenza virus on the course of COVID-19. The results indicate significant differences in the inflammatory and serological response between patients from both periods. The work addresses the important topic of the evolution of SARS-CoV-2 and the impact of mutations on the clinical course and immune response. The results may be important in further monitoring the effectiveness of vaccination and developing therapeutic strategies. However, I am missing some information to clarify and a few editorial corrections to increase the readability and clarity of the work, which I provide below:
- Although the authors mention that in the second cohort most patients were vaccinated, they do not provide precise information on the type and number of vaccine doses received. This data could help assess the impact of immunization on differences in the body's response.
- Lack of description of study limitations, e.g. small number of samples in the molecular study. HRM analysis includes only two samples from different periods, which may not be representative of the full range of SARS-CoV-2 variants present in the population at a given time.
- Although the work suggests that Omicron induces a strong inflammatory response, it does not analyze the long-term effects of COVID-19, e.g. occurrence of post-COVID syndrome.
- The authors show that in some cases there was an increase in the level of antibodies against influenza viruses, but they did not perform direct PCR tests for the presence of influenza virus, which makes the conclusions about the actual co-infection less precise.
- Figure 1 is very difficult to read, I propose to improve its quality and the distribution of names.
- The abstract seems too extensive in relation to the requirements of the journal
Author Response
The work by Yu. A. Desheva et al. entitled "The clinical and laboratory landscape of COVID-19 during the initial period of the pandemic and at the beginning of the Omicron era" is an interesting piece of literature. The article presents a retrospective clinical and laboratory analysis of hospitalized patients with COVID-19 in two different periods of the pandemic - the initial wave and at the turn of 2021, when the Omicron variant appeared. The authors examine differences in inflammatory biomarkers, immune response, and the impact of co-infection with influenza virus on the course of COVID-19. The results indicate significant differences in the inflammatory and serological response between patients from both periods. The work addresses the important topic of the evolution of SARS-CoV-2 and the impact of mutations on the clinical course and immune response. The results may be important in further monitoring the effectiveness of vaccination and developing therapeutic strategies.
Response:We thank the reviewer for the positive assessment of our work.
However, I am missing some information to clarify and a few editorial corrections to increase the readability and clarity of the work, which I provide below:
- Although the authors mention that in the second cohort most patients were vaccinated, they do not provide precise information on the type and number of vaccine doses received. This data could help assess the impact of immunization on differences in the body's response.
Response: We thank the reviewer for this remark. All patients received 2 doses of the S-protein-based vaccine SARS-CoV-2 (Sputnik-V). We indicated in the text.
- Lack of description of study limitations, e.g. small number of samples in the molecular study. HRM analysis includes only two samples from different periods, which may not be representative of the full range of SARS-CoV-2 variants present in the population at a given time.
Response: We thank the reviewer for this criticism. We have added the study limitation at the end of the manuscript.
- Although the work suggests that Omicron induces a strong inflammatory response, it does not analyze the long-term effects of COVID-19, e.g. occurrence of post-COVID syndrome.
Response: We thank the reviewer for this criticism. Indeed, our work does not analyze patients with post-Covid syndrome after Omicron. We are going to devote several of our future works to this issue.
- The authors show that in some cases there was an increase in the level of antibodies against influenza viruses, but they did not perform direct PCR tests for the presence of influenza virus, which makes the conclusions about the actual co-infection less precise.
Response: We thank the reviewer for this criticism. Indeed, our study did not include PCR tests for influenza viruses. However, a fourfold increase in antihemagglutinating antibodies to influenza viruses, which is considered the "gold standard" for serodiagnosis of influenza infection, cannot simply be ignored. Perhaps our publication will draw attention to the fact that in the case of severe COVID-19, a positive PCR test for influenza could provide a reason for prescribing specific antiviral therapy, which exists and works for influenza.
- Figure 1 is very difficult to read, I propose to improve its quality and the distribution of names.
Response: We thank the reviewer for this criticism. We fixed it.
- The abstract seems too extensive in relation to the requirements of the journal
Response: We thank the reviewer for this criticism. We have shortened the Abstract.